# A 10-Year Follow-Up of Ankle Syndesmotic Injuries: Prospective Comparison of Knotless Suture-Button Fixation and Syndesmotic Screw Fixation

**DOI:** 10.3390/jcm11092524

**Published:** 2022-04-30

**Authors:** Jan Niklas Altmeppen, Christian Colcuc, Christian Balser, Yves Gramlich, Alexander Klug, Oliver Neun, Sebastian Manegold, Reinhard Hoffmann, Sebastian Fischer

**Affiliations:** 1Department for Trauma and Orthopaedic Surgery, Berufsgenossenschaftliche Unfallklinik Frankfurt am Main, 60389 Frankfurt, Germany; jan-niklas.altmeppen@bgu-frankfurt.de (J.N.A.); christianbalser@hotmail.de (C.B.); yves.gramlich@bgu-frankfurt.de (Y.G.); alexander.klug@bgu-frankfurt.de (A.K.); reinhard.hoffmann@bgu-frankfurt.de (R.H.); 2Department of Trauma and Orthopaedic Surgery, Evangelical Hospital Bethel Bielefeld, 33611 Bielefeld, Germany; christian.colcuc@web.de; 3Department of Foot and Ankle Surgery, Berufsgenossenschaftliche Unfallklinik Frankfurt am Main, 60389 Frankfurt, Germany; oliver.neun@bgu-frankfurt.de (O.N.); sebastian.manegold@bgu-frankfurt.dee (S.M.)

**Keywords:** syndesmotic injury, syndesmotic reduction, suture-button device, syndesmotic screw fixation

## Abstract

Background: Acute syndesmosis injury (ASI) is an indication for surgical stabilization if instability is confirmed. In recent years, fixation using the knotless suture-button (SB) device has become increasingly established as an alternative to set screw fixation (SF). This study directly compared the clinical long-term results after prospective randomized inclusion. Materials and Methods: Between 2011 and 2012, 62 patients with ASI were enrolled in a prospective, randomized, and monocentric study. Forty-one patients were available for a 10-year follow-up ((31 males and 10 females), including 21 treated with SB (mean age 44.4 years), and 20 with SF (mean age 47.2 years)). In addition to comparing the demographic data and syndesmosis injury etiology, follow-up assessed the Olerud–Molander Ankle Score (OMAS) and FADI-Score (Foot and Ankle Disability Index Score) with subscales for activities of daily living (ADL) and sports activity. Results: The mean OMAS was 95.98 points (SB: 98.81, SF: 93.00), the mean FADI ADL was 97.58 points (SB: 99.22, SF: 95.86), and the mean FADI Sport was 94.14 points (SB: 97.03, SF: 91.10). None of the measurements differed significantly between the groups (*p* > 0.05). No clinical suspicion of chronic instability remained in any of the patients, regardless of treatment. Conclusions: The short-term results showed that athletes in particular benefit from SB fixation due to their significantly faster return to sports activities. However, the available long-term results confirm a very good outcome in the clinical scores for both approaches. Chronic syndesmotic insufficiency was not suspected in any of the patients. Level of evidence: I, randomized controlled trial.

## 1. Introduction

An intact distal syndesmotic ligament complex is essential for ankle joint stability and function, whereby undiagnosed, delayed, and untreated syndesmotic injuries lead to pain and joint wear. The epidemiology of syndesmosis injury varies, ranging from distortion trauma in sports such as soccer, football, and lacrosse to complex stair falls “slip-and-fall” and traffic accidents [1,2]. In most cases, the pathomechanism is excessive dorsiflexion, eversion, or external rotation [3]. The incidence of syndesmotic injuries in the context of ankle sprains is reported to be 20–40%, and it can be as high as 20–100% in the context of typical Weber B, Weber C, or Maisonneuve fractures [4,5].

For both fractures and acute ligamentous injuries, restoring a congruent articulating ankle with stable syndesmosis is essential to prevent post-traumatic osteoarthritis [6,7,8]. Surgical treatment of a chronic instability is clearly more complex and rarely achieves the good clinical results of an adequately treated acute syndesmosis injury [9,10,11].

Conventional treatment of acute syndesmosis injury, regardless of osteosynthetic treatment of a possible present fracture of the medial or lateral malleolus, predominantly involved screw fixation (SF). However, knotless suture-button (SB) fixation alone or with SF has also become an established procedure.

Clinical studies have already shown that use of the SB system in patients with an acute rupture of the distal tibiofibular syndesmosis can facilitate an earlier return to work and similar functional outcomes as measured by clinical scores [12]. Despite this evidence, there is still no uniform consensus for or against either treatment option [13,14].

The aim of this prospective randomized trial was to compare syndesmotic SF with knotless SB fixation in terms of clinical outcomes and complication rates. Since SB was already shown to hasten the return to sports and work, this investigation was designed to contrast long-term clinical outcomes [12]. The advantages and disadvantages of both reconstruction procedures were directly compared to make treatment recommendations.

## 2. Materials and Methods

### 2.1. Population

Between 2011 and 2013, 62 patients with acute syndesmotic injury were included in the monocentric prospective randomized study (Figure 1). The diagnosis of acute distal tibiofibular syndesmotic injury was always made on the basis of clinical examination with the dorsiflexion-external rotation test and the squeeze test under fluoroscopy, and plain radiographs were considered for enrollment. If an ankle fracture was initially detected, distal tibiofibular syndesmosis stability was checked intraoperatively.

The first group included SB fixation (Knotless TightRope^®^, Arthrex, Inc., Naples, FL, USA) and the second group included conventional 3.5 mm SF. After a mean follow-up of 118.34 months (approx. 10 years), we were able to evaluate the results of 41 patients (31 males and 10 females; SB: *n* = 21, SF: *n* = 20; mean age 45.78 years [SB: 44.43, SF: 47.20]) (Figure 1). The demographics of both groups were comparable (Table 1). All procedures were performed in accordance with the 1964 Helsinki Declaration and its later amendments or comparable ethics standards. The ethics committee of the institutional review board approved this study (FF 129/2011).

### 2.2. Inclusion and Exclusion Criteria

The inclusion criteria were a minimum age of 18 years and the presence of an acute isolated syndesmotic injury alone or in combination with an ankle fracture with syndesmotic injury. Radiographic findings included increased tibiofibular clear space and decreased tibiofibular overlap. If fluoroscopic examination with the dorsiflexion external rotation test showed a >2-mm difference between the affected and uninjured sides, the syndesmosis injury was defined as unstable [15]. Only patients who underwent surgery at the study center were included. Written informed consent was required prior to study participation.

Exclusion criteria included previous ankle surgery on the affected side or defined gait abnormalities due to previous rheumatic or neuromuscular disease. Patients on permanent pain therapy were also excluded.

### 2.3. Surgical Procedure

All surgeries were performed by the three surgeons who were well acquainted with both procedures. The operations were always performed under general anesthesia. All fractures were stabilized before insertion of the knotless SB device or 3.5-mm transosseous syndesmosis screw, and intraoperative stability was assessed performing the hook test under fluoroscopy.

Open reduction of the fibula was performed at the distal tibiofibular joint under direct vision. Correct setting of the syndesmosis reduction was further confirmed using Arcadis Orbic 3D fluoroscope three-dimensional (3D) imaging (Arcadis, Amsterdam, The Netherlands). The syndesmosis was then fixed depending on the randomization outcome: either with the SB device (Figure 2a,b) or according to the standard principles of the Association for the Study of Internal Fixation (AO-ASIF) using a 3.5-mm transosseous syndesmotic screw (Figure 3a,b) purchasing three cortices. An envelope with the group affiliation was opened just before the operation started.

Maisonneuve fractures were fixed with two of each implant. For SB fixation, a 4.0-cm quadricortical hole was drilled approximately 3.5–4.0 cm above the ankle joint line. The SB system was subsequently tightened and the syndesmosis was readapted using an absorbable suture.

### 2.4. Rehabilitation Protocol

On the second postoperative day, standard X-ray controls of the ankle were performed in two planes. All patients were immobilized in an orthotic boot (VACOped^®^ OPED Medical Inc., Buford, GA, USA) for 7 weeks postoperatively and were advised to use crutches. In case of an osteosynthetic-treated fracture, radiographic assessment was performed 6 weeks postoperatively to monitor fracture healing.

The VACOped^®^ was removed 7 weeks postoperatively and replaced with a stabilizing ankle brace. For patients in the SF group, the screw was removed as a follow-up outpatient procedure. The patient was then allowed pain-adapted weight-bearing and was instructed in proprioceptive training under physical therapy guidance.

### 2.5. Assessment Methods

All patients were followed up by an independent surgeon who was not blinded to the surgery type. This was followed by review of radiographs before resumption of full weight-bearing, and at 6 and 12 months postoperatively to exclude degenerative changes.

Clinical outcomes were assessed using the American Orthopaedic Foot and Ankle Society (AOFAS) score, the Olerud–Molander score (OMAS), and the Foot and Ankle Disability Index (FADI) at 1 year postoperatively. The latter two scores were collected again at approximate 9.5 years postoperatively. The times from surgery to return to work and full athletic activity were also documented.

The short-term results were collected during a clinical examination. Due to the pandemic situation, the long-term results (OMAS, FADI) were mainly collected by telephone. For this reason, a new survey of the AOFAS score had to be dispensed with.

### 2.6. Statistical Analysis

All calculations were performed by a professional statistician using SPSS software (SPSS Statistics for Windows, version 21.0, IBM Corp., Armonk, NY, USA) and BIAS software (Biometric Analysis of Samples Software, version 8.4.2, BIAS Corp., Roswell, GA, USA).

Demographic data were compared using means and proportions between the two groups. Clinical scores were compared between the two groups using the Mann–Whitney U test for independent samples. Means were calculated for clinical outcomes, and Student’s *t*-tests were used to compare these scores within each group. Spearman rank correlation test and bivariate analysis were performed to adjust for potential confounding factors such as age, body mass index (BMI), surgical procedure, and injury type to identify variables affecting clinical outcomes and time to return to sport. Before starting the present study, the following calculation was performed. A priori analysis of the required sample sizes using the Mann–Whitney U test was performed with an effect size of d = 0.8 (large effect), α = 0.05, and power (1 − β) of 0.80, resulting in a sample size of 62 participants (*n* = 31 per group). All calculations were performed according to per-protocol analysis. Differences were considered significant at *p* ≤ 0.05.

## 3. Results

Both groups had comparable demographics, including the same risk profiles of pre-existing disease, BMI, and nicotine abuse (Table 1). The median follow-up time for all patients was 118.34 months (range 118.00–128.00). The mean time to return to work, 10 weeks (SB: 9, SF: 11), did not differ significantly between the groups.

At 12 months postoperatively, the mean AOFAS scores were 91.00 points for SB and 91.00 for SF. For all patients, the mean OMAS and FADI ADL scores improved from 90.00 points and 95 ± 8 points to means of 95.98 and 97.58 points, respectively, from 1 year postoperatively to the last examination approximately 9 years postoperatively.

A significant difference was observed in the time to return to sport. It was 14 weeks in the SB group and 19 weeks in the SF group (*p* = 0.006, effect size d = 0.63). FADI Sport showed the most relevant difference over time, improving from a mean of 84.00 points to a mean of 94.14 points (SB: 97.03, SF: 91.10). However, differences over time as shown in Table 2 and Figure 4 were not significant (*p* = 0.254).

When subgrouped by injury mechanism, patients with isolated syndesmosis injury had the best outcomes in all scores. As shown in Table 3, Weber B fractures with concomitant syndesmosis rupture had the worst outcome. This tendency was observed within both groups, and the differences were not statistically significant. Due to the insufficient subgroup sizes, they are not presented separately.

The use of a Suture-button device or Syndesmotic screw fixation had no significant effect on the clinical outcome after Maisonneuve injury (*p* > 0.05).

While all screws were removed 7 weeks postoperatively, implant removal in the SB group occurred only nine times within the first 2 postoperative years, although one time occurred as early as 3 months after implantation due to local irritation at the tibial cortex. No other relevant complications were observed, and there were no infections requiring intervention.

## 4. Discussion

The most important finding of this study is that both the SB and SF procedures were sufficient treatments for acute syndesmotic injury. Equivalence, especially with regard to clinical scores, has been demonstrated several times in retrospective studies with a large numbers of cases and in prospective studies with comparatively short follow-up periods [12,16,17,18,19,20]. In vivo analyses with 3D computer imaging techniques confirmed comparable reduction following treatment of syndesmotic rupture with SB or SF [16,21]. Randomized controlled trials and meta-analyses with comparable numbers of patients reported clinical results at a similarly high level; a limitation is the mostly short follow-ups [20,22,23].

The present study presents good to excellent clinical results in short-, mid-, and long-term results for the entire study population. All scores continued to improve over time, with FADI Sport gaining the most points proportionally. The short postoperative results of the present study already tended to be better for the SB group (Figure 4). Of note is the significantly faster return to sport at 14 weeks (SB) compared to 19 weeks (SF). The findings from the present study, which only included recreational athletes, are consistent with the experience of D’Hooghe et al. in professional sports [24]. Our long-term results confirm an improvement in all scores for all patients over time, with better results for patients treated with SB. The clinical results were on comparably high levels (AOFAS > 90 points, OMAS > 90 points), as previously they demonstrated using other studies that exclusively included isolated syndesmosis injuries [25,26,27,28].

Although individual biomechanical studies still question the equivalence of SB and SF with regard to stability, two studies found no clinically relevant differences [29,30]. However, others have discussed whether a less rigid fixation could facilitate a more physiological healing of the syndesmosis injury [31,32].

Both the SB and SF surgical procedures were in accordance with the accepted literature, so all cortical screws were inserted tricortically. We were previously unable to identify any advantage of quadricortical screw fixation [33]. The same applies to SB fixation. Parker et al. showed that the three possible SB implantations of “single, parallel, and divergent” exhibit comparable stabilities in their biomechanical study [34]. This fact was confirmed in clinical studies by Kurtoglu et al., who saw no evidence for implantation of more than one SB device to treat acute syndesmotic rupture [35].

The clinical superiority of the SB is attributed to the absence of a second metal removal procedure before weight bearing, among other factors. Again, a consensus seems to be emerging that leaving the syndesmotic screw in place is adequate. Fractured screws are not a complication and should not be routinely removed unless they cause symptoms [36]. Standardization of this procedure could help confirm the superiority of the SB in the future. There may be an associated improved cost efficiency in the absence of screw removal, but that issue was not investigated in the present study [14,37].

Based on the available data of the present study, it is difficult to differentiate whether, in the case of fibula plate fixation and SB fixation, subsequent implant removal was performed in isolation because of interfering plate osteosynthesis or an interfering SB. This scenario occurred six times. A similar proportion of patients without fibula plate fixation complained of local irritation due to SB and desired removal. The alleviation of pain and discomfort after implant removal reported by Kim et al. is consistent with the authors’ experience [38]. SB removal is a minimally invasive procedure with few complications that can usually be performed on an outpatient basis.

The negative influence of tobacco use with regard to wound and ligament healing and pain sensation is well known [39,40,41]. However, this negative effect could not be demonstrated in the present study.

Posttraumatic arthrosis of the ankle joint is a feared complication after acute syndesmosis rupture, especially in cases with delayed diagnosis and inadequate surgical or conservative therapy [42,43,44]. Lehtola et al. exclusively followed-up patients who sustained Weber C fractures in combination with syndesmotic injuries. After a mean follow-up of 7 years, they confirmed mild to moderate ankle osteoarthritis in the syndesmotic screw and subture button groups, 9 of 16 and 11 of 13 patients, respectively, with a mean OMAS of 83 points [45]. In the present study, we found a significantly better clinical outcome (OMAS of 95 points), but study-related radiographic examinations were not performed. The good clinical outcome suggested that there was no relevant osteoarthritis development.

A particular strength of the prospective randomized study is the follow-up of 10 years and the good comparability of the demographic data of all patients.

Our results should be considered in the context of some limitations. The follow-up rate was greater than assumed at the time. Therefore, the statistical power might have decreased. Current radiographs are not available for all patients. For reasons of radiation protection, radiographic imaging was not performed in cases with good to excellent clinical results. Early stages of posttraumatic osteoarthritis could therefore have gone undetected.

## 5. Conclusions

The short-term results demonstrated that athletes in particular benefit from SB fixation due to the significantly faster return to sports. The available long-term results confirm very good clinical scores for both the SB and syndesmotic SF groups. Chronic syndesmotic insufficiency was not observed in any of the patients in this study.

## Figures and Tables

**Figure 1 jcm-11-02524-f001:**
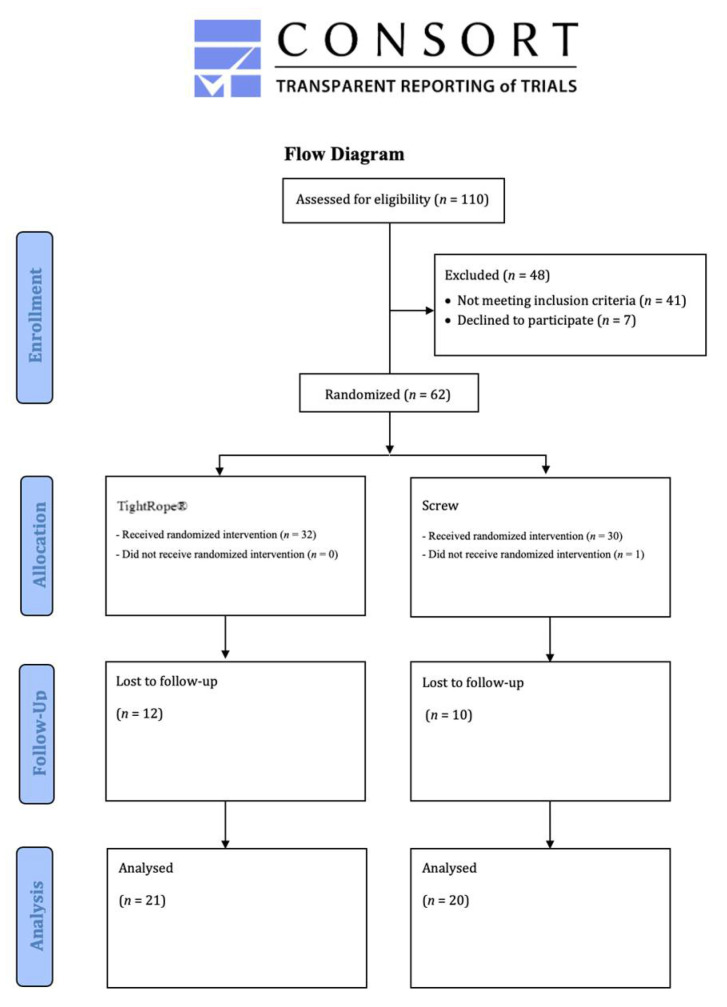
Study flow chart.

**Figure 2 jcm-11-02524-f002:**
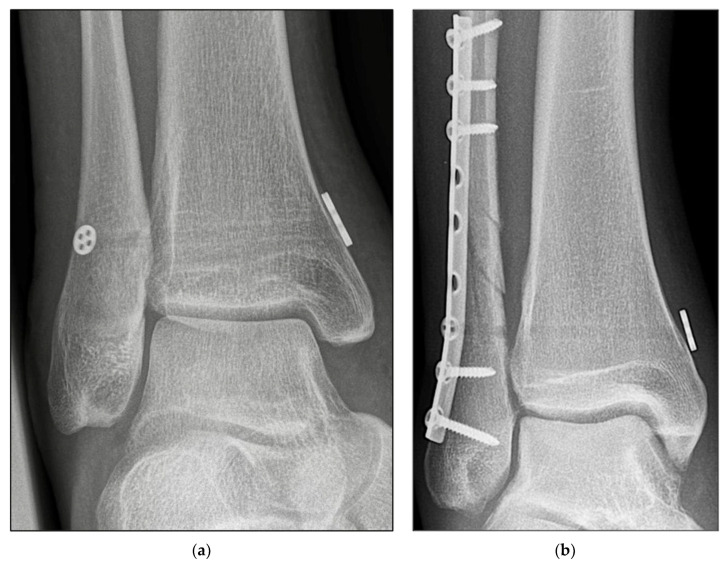
Radiographic imaging. (**a**) Anteroposterior view of isolated Knotless Suture-button fixation (TightRope^®^), left ankle, male 41 years. (**b**) Anteroposterior view of Plate fixation of Weber C fracture and Knotless Suture-button fixation (TightRope^®^), right ankle, female 52 years.

**Figure 3 jcm-11-02524-f003:**
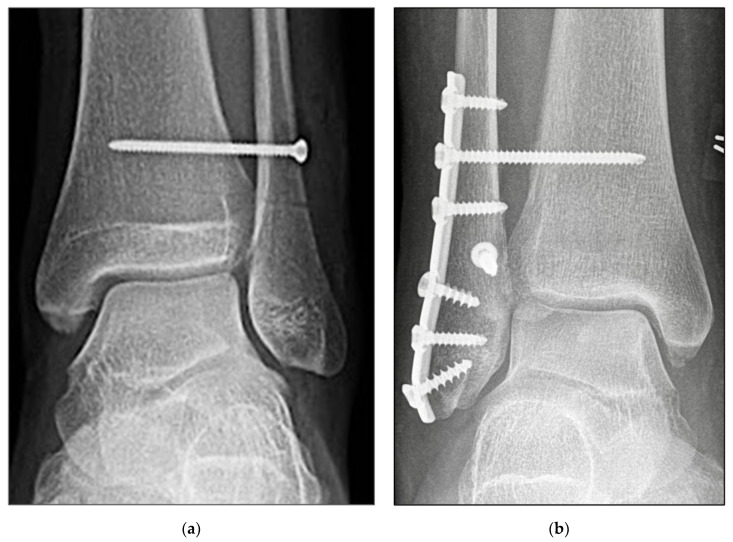
Radiographic imaging. (**a**) Anteroposterior view of isolated Syndesmotic screw fixation, left ankle, male 45 years. (**b**) Anteroposterior view of Plate fixation of Weber B fracture and Syndesmotic screw fixation, right ankle, male 48 years.

**Figure 4 jcm-11-02524-f004:**
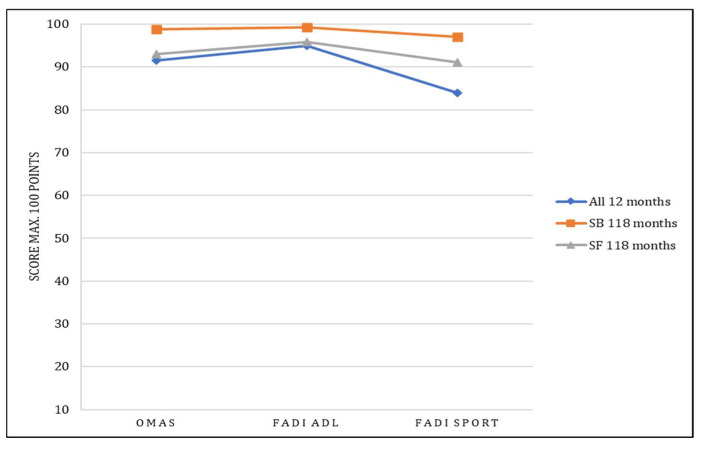
Outcome over time. Mid- and long-term results at 1 and 10 years, respectively; ADL, activities of daily living; FADI, Foot and Ankle Disability Index; OMAS, Olerud–Molander Ankle Score.

**Table 1 jcm-11-02524-t001:** Patient characteristics at the time of follow-up.

Characteristic		SB (*n* = 21)	SF (*n* = 20)	All (*n* = 41)	*p*
Age, years					
	Mean	44.43	47.20	45.78	0.458
	Minimum	27.00	25.00	25.00	
	Maximum	61.00	77.00	77.00	
BMI, kg/m^2^					
	Mean	27.810	28.05	27.92	0.880
	Minimum	21.600	19.50	19.50	
	Maximum	39.100	37.60	39.10	
Sex, *n* (%)					
	Male	16 (76.19)	15 (75)	31 (75.61)	0.931
	Female	5 (23.81)	5 (25)	10 (24.39)	
Affected side, *n* (%)					
	Left	11 (52.38)	14 (70.00)	25 (60.98)	0.259
	Right	10 (47.61)	6 (30.00)	16 (39.02)	
Smoker, *n* (%)					
	Yes	5 (23.81)	3 (15.00)	8 (19.15)	0.489
	No	16 (76.19)	17 (85.00)	33 (80.49)	
Pre-existing conditions, *n* (%)					
	Metabolic syndrome-associated	7 (33.33)	3 (15.00)	10 (24.39)	0.633
	Others	4 (19.07)	9 (45.00)	13 (31.71)	
	None	10 (47.60)	8 (40.00)	18 (43.90)	
Injury classification, *n* (%)					
	Weber B	2 (9.53)	4 (20.00)	6 (14.63)	0.338
	Weber C	2 (9.53)	3 (15.00)	5 (12.19)	
	Isolated ^a^	11 (52.38)	8 (40.00)	19 (46.34)	
	Maisonneuve	6 (28.57)	5 (25.00)	11 (26.83)	

BMI, body mass index; SB, suture-button; SF, screw fixation; *p*, *p*-value; ^a^ isolated syndesmotic injury.

**Table 2 jcm-11-02524-t002:** Outcome according to treatment.

Measurements	SB (*n* = 21)	SF (*n* = 20)	All (*n* = 41)	*p*
Follow-up in months				
Mean	116.52	120.25	118.34	0.056
Range	105.00–124.00	107.00–128.00	105.00–128.00	
OMAS				
Mean	98.81	93.00	95.98	0.101
SEM	0.59	3.49	1.77	
Minimum	90.00	30.00	30.00	
Maximum	100.00	100.00	100.00	
FADI Score ADL				
Mean	99.22	95.86	97.58	0.154
SEM	0.36	2.34	1.17	
Minimum	94.20	54.80	54.80	
Maximum	100.00	100.00	100.00	
FADI Score Sport				
Mean	97.03	91.10	94.14	0.254
SEM	1.41	4.99	2.55	
Minimum	78.10	15.60	15.60	
Maximum	100.00	100.00	100.00	

ADL, activities of daily living; FADI, Foot and Ankle Disability Index; OMAS, Olerud–Molander Ankle Score; SB, suture-button; SF, screw fixation; SEM, standard error of the mean.

**Table 3 jcm-11-02524-t003:** Patient outcomes according to injury mechanism (*n* = 41).

Measurements	Isolated Syndesmotic Injury	Weber B and Syndesmotic Injury	Weber C and Syndesmotic Injury	Maisonneuve and Syndesmotic Injury	*p*
OMAS					
Mean (range)	97.90	85.00	98.00	97.73	0.079
SEM	0.88	11.18	1.23	1.41	
Minimum	90.00	30.00	95.00	85.00	
Maximum	100.00	100.00	100.00	100.00	
FADI Score ADL					
Mean	99.089	91.017	98.640	98.073	0.135
SEM	0.371	7.296	1.127	1.560	
Minimum	95.200	54.800	94.200	82.700	
Maximum	100.000	100.000	100.000	100.000	
FADI Score Sport					
Mean	97.21	82.82	96.88	93.76	0.303
SEM	1.37	13.79	3.12	5.35	
Minimum	78.10	15.60	84.40	40.60	
Maximum	100.00	100.00	100.00	100.00	

ADL, activities of daily living; FADI, Foot and Ankle Disability Index; OMAS, Olerud–Molander Ankle Score; SEM, standard error of the mean.

## Data Availability

All data intended for publication are included in the manuscript.

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
