# Peer review of "A 10-Year Follow-Up of Ankle Syndesmotic Injuries: Prospective Comparison of Knotless Suture-Button Fixation and Syndesmotic Screw Fixation"

_jcm, 2022, doi:10.3390/jcm11092524_

Round 1

Reviewer 1 Report

Congratulations for the good work.

Overall well designed and written.

Few comments:

Did you perform a power calculation of the sample size? If yes add to population part if not add to limitaions.

What was the type of randomisation used? Please add to methods

How was the feedback AOFAS, FADI collected at 9 years? F2F and clinically , phone etc. Please add. 

You have used 2 SB or SF in Maisonneuve  injuries, was there any significant difference between those and single implant fixation? 

In the surgical technique,  what was the test used intraoperative to assess syndesmotic stability,  please add.

Author Response

Dear Reviewer,

Thank you very much for your helpful comments. The suggested changes have been fully edited. 
Please see the attachment.

Reviewer 2 Report

Comments to the authors of the manuscript ID jcm-1697836 entitled “A 10-Year Follow-up of Ankle Syndesmotic Injuries: Prospective Comparison of Knotless Suture-button Fixation and Syndesmotic Screw Fixation”. This is a well structured manuscript about the variability in the postoperative syndesmosis treatment using knotless suture button versus screw fixation. Greetings to the authors for the research and minor changes are required.

0.- Abstract: Is well structured and adapted to the requirements

1.- Introduction: Authors must add more previous research. The introduction is insufficient.

2.- Material and methods.

Figure 1: please modified the study flow chart. Is too big.

Table 1: please add abbreviations word and add the p value in abbreviations

Material and methods are well describe

3.- Results.

Are well explained

4.- Discussion

Must be improve. Please add references about the influence of the nicotine. Authors add smokers but does not explain the effects.

5.- Limitations. Add in the discussion.

6.- Conclusion_ Ok

Author Response

(The authors gave the same response as above.)

Round 2

Reviewer 2 Report

Thanks to the authors for realize the changes in manuscript required. No more changes are required.